# A Comprehensive Analysis and Anti-Cancer Activities of Quercetin in ROS-Mediated Cancer and Cancer Stem Cells

**DOI:** 10.3390/ijms231911746

**Published:** 2022-10-04

**Authors:** Partha Biswas, Dipta Dey, Polash Kumar Biswas, Tanjim Ishraq Rahaman, Shuvo Saha, Anwar Parvez, Dhrubo Ahmed Khan, Nusrat Jahan Lily, Konka Saha, Md Sohel, Mohammad Mehedi Hasan, Salauddin Al Azad, Shabana Bibi, Md. Nazmul Hasan, Mohammed Rahmatullah, Jaemoo Chun, Md. Ataur Rahman, Bonglee Kim

**Affiliations:** 1Laboratory of Pharmaceutical Biotechnology and Bioinformatics, Department of Genetic Engineering and Biotechnology, Jashore University of Science and Technology, Jashore 7408, Bangladesh; 2ABEx Bio-Research Center, East Azampur, Dhaka 1230, Bangladesh; 3Biochemistry and Molecular Biology Department, Life Science Faculty, Bangabandhu Sheikh Mujibur Rahman Science and Technology University, Gopalgonj 8100, Bangladesh; 4Department of Stem Cell Regenerative Biotechnology and Institute of Advanced Regenerative Science, Konkuk University, 120 Neungdong-ro, Gwangjin-gu, Seoul 05029, Korea; 5Division of Biological Sciences, University of Montana, Missoula, MT 59812, USA; 6Department of Biotechnology and Genetic Engineering, Faculty of Life Science, Bangabandhu Sheikh Mujibur Rahman Science and Technology University, Gopalganj 8100, Bangladesh; 7Department of Pharmacy, Faculty of Allied Health Sciences, Daffodil International University, Dhaka 1207, Bangladesh; 8Department of Microbiology, Stamford University, Dhaka 1217, Bangladesh; 9Shaheed Taj Uddin Ahmad Medical College, Gazipu 1712, Bangladesh; 10Department of Biochemistry and Molecular Biology, Faculty of life Science, Mawlana Bhashani Science and Technology University, Santosh, Tangail 1902, Bangladesh; 11School of Biotechnology, Jiangnan University, 1800, Lihu Avenue, Wuxi 214122, China; 12Department of Bioscience, Shifa Tameer-e-Millat University, Islamabad 44000, Pakistan; 13Yunnan Herbal Laboratory, College of Ecology and Environmental Sciences, Yunnan University, Kunming 650091, China; 14Department of Biotechnology Genetic Engineering, University of Development Alternative, Lalmatia, Dhaka 1207, Bangladesh; 15KM Convergence Research Division, Korea Institute of Oriental Medicine, Daejeon 34054, Korea; 16Global Biotechnology Biomedical Research Network (GBBRN), Department of Biotechnology and Genetic Engineering, Faculty of Biological Sciences, Islamic University, Kushtia 7003, Bangladesh; 17Department of Pathology, College of Korean Medicine, Kyung Hee University, Seoul 02447, Korea; 18Korean Medicine-Based Drug Repositioning Cancer Research Center, College of Korean Medicine, Kyung Hee University, Seoul 02447, Korea

**Keywords:** ROS, REDOX imbalance, carcinogenesis, malignant cells, quercetin, cancer stem cells

## Abstract

Reactive oxygen species (ROS) induce carcinogenesis by causing genetic mutations, activating oncogenes, and increasing oxidative stress, all of which affect cell proliferation, survival, and apoptosis. When compared to normal cells, cancer cells have higher levels of ROS, and they are responsible for the maintenance of the cancer phenotype; this unique feature in cancer cells may, therefore, be exploited for targeted therapy. Quercetin (QC), a plant-derived bioflavonoid, is known for its ROS scavenging properties and was recently discovered to have various antitumor properties in a variety of solid tumors. Adaptive stress responses may be induced by persistent ROS stress, allowing cancer cells to survive with high levels of ROS while maintaining cellular viability. However, large amounts of ROS make cancer cells extremely susceptible to quercetin, one of the most available dietary flavonoids. Because of the molecular and metabolic distinctions between malignant and normal cells, targeting ROS metabolism might help overcome medication resistance and achieve therapeutic selectivity while having little or no effect on normal cells. The powerful bioactivity and modulatory role of quercetin has prompted extensive research into the chemical, which has identified a number of pathways that potentially work together to prevent cancer, alongside, QC has a great number of evidences to use as a therapeutic agent in cancer stem cells. This current study has broadly demonstrated the function-mechanistic relationship of quercetin and how it regulates ROS generation to kill cancer and cancer stem cells. Here, we have revealed the regulation and production of ROS in normal cells and cancer cells with a certain signaling mechanism. We demonstrated the specific molecular mechanisms of quercetin including MAPK/ERK1/2, p53, JAK/STAT and TRAIL, AMPKα1/ASK1/p38, RAGE/PI3K/AKT/mTOR axis, HMGB1 and NF-κB, Nrf2-induced signaling pathways and certain cell cycle arrest in cancer cell death, and how they regulate the specific cancer signaling pathways as long-searched cancer therapeutics.

## 1. Introduction

The term “ROS (Reactive Oxygen Species)” refers to radicals and ions that contain an unpaired numbered electron in its outmost electron field which are highly reactive metabolic byproducts that can have both harmful and useful effects within the cell [1,2]. Basically, free oxygen radicals and non-radical ROS are the two main sorts of ROS in which the free oxygen radicals are including numerous superoxide (O_2_), besides, the non-radical ROS include the H_2_O_2_ (hydrogen peroxide) [3]. Moreover, the most widely studied ROS in cancer killing are given here including hydrogen peroxide, hydroxyl radicals, and superoxide. Cellular ROS functions in signaling cascades as secondary messengers crucial for the general physiological performances known as differentiation and development [4,5]. However, excessive ROS production can also damage the body’s biological substances such as DNA, carbohydrates, lipids, as well as proteins, leading to loss of cell integrity and subsequent cell pathology [6,7]. Clinical diseases such as inflammation, atherosclerosis, angiogenesis, aging, and cancer are influenced by abnormal ROS regulation [8]. Of late, it is now recognized that ROS promote metastasis, angiogenesis, and tumorigenesis [9], while excessive activation of ROS causes cell death [10]. Various research findings demonstrated that cancer cells or rapidly proliferating cells have higher metabolism rates as well as mitochondrial dysfunction along with being more susceptible under oxidizing conditions than healthy cells [11,12]. It has been extensively studied how elevated ROS levels correlate with oncogenic activity. For example, compared to quiescent, untransformed cells, different hematopoietic cell lines transformed with BCR/ABL result in higher ROS levels [13]. Consequently, elevated ROS levels within cancerous cells can easily exceed their oxidative stress conditions, and most importantly such mechanisms directly involve the induction of cell death [11,12,14,15]. 

The discovery or development of chemicals that can kill malformed or cancer cells, without being toxic or harming their healthy counterparts, is of utmost significance, and has attracted the attention of scientists all over the world [16,17,18,19]. Some plant-derived substances or phytochemicals have been found to be highly effective anticancer therapies in recent years, in addition also beneficial against a variety of other disorders [20,21,22,23,24,25,26]. Throughout history, plant extracts and their purified active components have long been the foundation of cancer chemotherapies [27,28,29,30]. It is estimated that over 70 percent of anticancer chemicals are natural products or their derivatives [31,32,33,34,35]. Flavonoids have significant antioxidant properties by regulating the activity of several detoxifying enzymes such as cyclooxygenases, lipoxygenases, and inducible nitric oxide synthase, as well as decreasing ROS formation [36,37,38,39,40,41]. Flavonoids have been shown to have anti-carcinogenic properties in cell culture and animal models, owing to their antioxidant activity and ability to alter proteins that are involved in cell proliferation, and cell death mechanisms [8,42,43,44]. Quercetin (QC), which bears the chemical name “3,3′,4′,5,7-pentahydroxyflavone”, is one of the most plentiful flavonoids under the flavonol group which has diverse biological activity for instant- antioxidant, anti-inflammatory, and antitumor activity, and is widely studied in several cancer models as a chemotherapeutic option [45,46,47]. It can disrupt ROS metabolism and trigger subsequent apoptosis, conversely, significantly raising intracellular ROS levels by forming QC radicals (QC-O•) after peroxidase-catalyzed oxidation to scavenge harmful reactive peroxyl radicals [48]. Indeed, QC has been shown to inhibit the progression of cancers such as lung, cervical, prostate, breast, and colon. Current research articles proposed that QC directly involves inducing apoptosis and/or the cell cycle arrest process, and also inhibits the propagation of rapidly proliferating cells [49]. Notably, therapy with QC results in cell cycle arrest, mainly G2/M or G1 arrest in numerous cell types [50]. By directly reducing the intracellular pool of glutathione (GSH), QC can influence ROS metabolism [51,52]. Given its high toxicity for cancer cells and its very limited effects on normal, non-transformed cells, research on QC as prospective chemopreventers or for chemotherapy is becoming increasingly important [53,54]. Additionally, quercetin-dependent cell death may arise due to an instability of mitochondria and/or microtubules, the activation of stress proteins, triggering the caspases-mediated pathways, and the release of cytochrome c [55]. However, QC is a powerful antioxidant due to its ability to oxidize metals ions, inhibit xanthine oxidase, scavenge oxygen free radicals, and lipid peroxidation in vitro [56]; it triggers apoptosis via the mitochondrial pathway, through the loss of mitochondrial membrane potential, following the release of cytochrome c from mitochondria to the cytosol, and the ultimate activation of caspases, such as caspase-3 and caspase-7, all initiate apoptosis via the mitochondrial route [49,57,58]. Besides, non-transformed cells are less susceptible to quercetin-induced cell death as well as a variety of GSH-depleting stimuli [59,60]. In human K562 chronic myeloid leukemia cell lines, PHICNQ, a QC derivative, promotes apoptosis by downregulating Hsp-70 [61,62]. In addition to its proapoptotic capacity, in squamous cell carcinoma, breast, lung, and hepatoma cancer cells, QC increases cell cycle arrest via regulating p21WAF1, cyclin B, and p27KIP1 [63]. Several in vivo investigations on animal models showed that it is effective in preventing an extensive number of cancers caused by powerful carcinogens such as benzo(a)pyrene, azoxymethane, and N-nitrosodiethylamine [64,65,66]. Furthermore, clinical studies of quercetin have been conducted, with no evidence of toxicity or adverse effects in the population [67]. In terms of safety studies, the nearly complete absence of adverse effects in preclinical studies (in terms of acute toxicity, in vitro and in vivo carcinogenicity), at least at the estimated dietary intake, makes QC a promising candidate for clinical applications in the treatment of a diverse number of cancers [68]. However, more extensive investigations on its actual mechanism, and activities in humans are required, as only a few clinical studies have evaluated QC as a potential chemotherapeutic.

Quercetin plays a crucial role in several signal transduction pathways. It directly activates the MAPK/ERK-mediated pathways, leading to the apoptosis process and the whole process was discovered within the A549 cell line [69]. Quercetin inhibited tumor cell proliferation by stimulating two different regulatory networks, p53 and NF-κB [70]. An investigation reported that in A549 human lung cancer cells expressing wild-type p53, cell growth arrest and apoptosis were accelerated by quercetin considerably [71]. Moreover, QC-mediated anti-inflammatory and anti-apoptotic properties play a key role in cancer prevention by modulating the TLR-2 (toll-like receptor-2) and JAK-2/STAT-3 pathways and significantly inhibit STAT-3 tyrosine phosphorylation within inflammatory cells [72]. Importantly, QC blocks different pathways and is also involved in controlling cancers. 

This current study has demonstrated the ROS regulation in normal cells, production of ROS in both cancer, and cancer stem cells, the potential role of ROS in cancer cell signaling, and quercetin’s role in an unbalanced ROS state in cancerous cells. Additionally, the role of QC-dependent apoptosis, cell death, and major potential molecular mechanisms (for example, MAPK/ERK1/2 pathways, p53 pathway, and AMPKα1/ASK1/p38 pathway) for quercetin-induced cancer and cancer stem cell death is evaluated, and lastly, the future directions and precise investigations that are required to make quercetin a safe and effective anticancer pharmaceutical product are highlighted.

## 2. Overview of Quercetin 

Quercetin (Figure 1) is a naturally occurring polyphenolic flavonoid, whose chemical name is 3,3′,4′,5,7 pentahydroxyflavone (C_15_H_10_O_7_), can be found in a wide range of fruits and vegetables such as capers, lovage, dill, and cilantro as well as apples and berries including chokeberries and cranberries [73]. 

Studies suggested that quercetin can be isolated from the phytochemical of several plant extracts in which quercetin can exist as an aglycone or its derivatives by adjoining with: (1) Quercetin glycoside composed as carbohydrate, (2) quercetin conjugated with lipids, (3) alcohol as quercetin ethers, and (4) quercetin sulfate as a sulfate group [74]. To evaluate the anticarcinogenic efficacy of quercetin, it is necessary to comprehend its bioavailability, intestinal absorption, and metabolic conversion rate [75]. It is abundantly clear that quercetin is metabolized and excreted out of the body through the urine in a short amount of time, and there is no evidence found to indicate that quercetin is stored in the tissues or body fluids [76]. Quercetin and its metabolites are essential because of their physiological effects with significant therapeutic potential and less side effects. In which, the antioxidant activity of this flavonoid is possibly the most important attribute that it possesses [77]. In addition, quercetin has been shown to be effective in the treatment of cancer, in regard to providing properties that are antiviral, antiallergic, and anti-inflammatory. Most notably, quercetin inhibits the growth of a number of different malignancies, including those that have a close relation to the lung, prostate, liver, breast, colon, and cervical tissues; and here such activities are mediated by a variety of mechanisms, including cellular signaling and the ability to block enzymes that activate carcinogens [78]. Due to the fact that quercetin binds to cellular receptors and proteins, it can inhibit the growth of cancerous cells [79,80]. Additionally, it was recently discovered that the antioxidant quercetin has synergistic benefits when paired with chemotherapeutic drugs such as cisplatin. Therefore, these effects may help conventional chemotherapy be more successful in treating cancer [81].

## 3. ROS Regulation within Cellular System 

Environmental influences such as tobacco, smoke, and ionizing radiation, as well as intracellular indicators such as the endoplasmic reticulum (ER), mitochondria, and peroxisomes regulate ROS production (Figure 2) [82]. Here, the endogenous ROS are usually generated in mitochondria via following the oxidative phosphorylation process. Contrastingly, superoxide anions (SA) are created in the I and III transportation chain, which is located in an inner part of the mitochondrial mucosa, and the SOD-superoxide dismutase converts the SA into H_2_O_2_, and catalase (CAT) also transforms H_2_O_2_ into water [83]. ROS production is also possible with diverse enzymes, which are mainly intracellular enzymes including lipoxygenases, NADPH oxidase, as well as xanthine oxidase [84]. The enzymatic antioxidants SOD, GPX, and CAT are responsible for maintaining intracellular redox homeostasis; besides, the nonenzymatic antioxidants namely ascorbic acid and GSH also displayed an indispensable role [85]. Apart from such antioxidants, the transcription factor activator protein erythroid-related factor-2 simply known as “Nrf-2”, also followed a significant role under oxidative stress conditions at the same time regulating the oxidation-mediated pathways. Interestingly, Nrf-2 activation is facilitated by inhibiting its negative regulator, Keap1, resulting in Nrf-2 nuclear translocation [86]. This results in the production and activation of enzymes namely GPX, CAT, heme oxygenase 1 (HO-1), peroxiredoxin (PRX), and fostering the REDOX balance [86].

In addition, the aggregation of excessive production of ROS can be toxic, but specific amounts of ROS are required for an effective signaling process. It is critical for cells to maintain strict control over their mitochondrial ROS (mROS) levels [87]. Antioxidant scavenging proteins also take part in a significant role in regulating the formation of mROS within cellular compartments. The formation of mROS is poorly understood in vivo, hence research on isolated mitochondria and cells in vitro has provided valuable insight into its regulation [88]. Numerous factors have been discovered to influence mROS formation, including the amount of intracellular O_2_ level, the frequency of electron delivery to the electron transport chain (ETC), the concentrations of a specific electron transport system, and the REDOX level within the ETC [89]. mROS is regulated by the mitochondrial membrane potential (MMP), which is a chemical and electrical gradient that forms across the mitochondrial inner membrane as protons are pushed into the inter-membrane region by different subunits such as I/II/III/IV of the ETC. The presence of enhanced MMP has been linked to a rise in mitochondrial ROS [90]. Also of note, loss of mROS is associated with a reduction in mROS production, which may be recovered with the restoration of MMP, demonstrating that it is both essential and sufficient for mROS formation [91].

## 4. Enzymatic Mechanism of ROS Production in Cancer and Cancer Stem Cell

In mammalian cells, mitochondrial ETC is the rudimentary source of ROS, where electrons leak from the byproducts. Moreover, nuclear or mitochondrial gene mutations that encode ETC components can block electron transmission resulting in electron leakage [92]. Afterward, the electrons are collected by O_2_, which form anionic oxygen (O_2_^−^) and are typically converted to H_2_O_2_ with several enzymatic actions: copper/zinc-derived cytosolic superoxide dismutase1 (Cu/Zn-SOD-1), manganese-mediated mitochondrial superoxide dismutase (Mn-SOD, SOD2), and cytoplasmic SOD-3 [93]. In the same way, the H_2_O_2_ diffuses into the nucleus and oxidizes chromosomal DNA, and the Fenton reaction can also be used to catalyze the release of HO• from H_2_O_2_ by the attendance of Fe^2+^/Cu^2+^ ions [11]. Additionally, O_2_^−^ can be generated in macrophages and cancer cells via a coupled reaction by the membrane-localized NAD(P)H oxidase complex (NOX), which comprises NOX 1/2/3/4, cytochrome c oxidase, cyclo-oxygenase, and cytoplasmic reticulum-associated xanthine oxidase-XO. Here, the NO• is formed from the arginine through a catalytic reaction, and the reaction is mediated via the nitric oxide synthase (NOS) enzyme, which is found in various tissues and has several tissue-mediated transporters known as endothelial NOS, mitochondrial NOS, inducible NOS, and neuronal NOS [94]. By reacting with O_2_, NO• is easily translated to ONOO^−^, a reactive nitrogen species (RNS) that is frequently classified as ROS due to its oxidative nature. Here, ONOO- can exacerbate tyrosine residue nitration, which inhibits protein activity by preventing gene expression and adenylation of these specific tyrosine residues [95]. For instance, HIF-1 protein accumulates when ROS are directly applied to the human prostate cancer stem cell line at concentrations of 0.5 mM H_2_O_2_ and 100 M menadione, which inhibits its degradation and increases its transcriptional activity in an AMPK-dependent way [96]. During oxidative stress, AMPK suppression boosts HIF-1 proteasomal degradation. As a result of H_2_O_2_, Janus kinase 2 and the c-Jun N terminal kinase pathways regulate AMPK’s regulation of HIF-1. When taken as a whole, AMPK can be seen as a key factor determining how HIF-1 responds to H_2_O_2_ and may be involved in complex HIF-1 regulatory mechanisms [97,98]. According to research studies, while under oxidative stress, cancer cells switch off the glycolytic route in favor of increasing glucose flux through the PPP, which produces more NADPH for antioxidant defense [99]. When exposed to ionizing radiation, chemotherapy, or oxidative stress, raises ROS levels and causes an adaptive response by enhancing the activity of the pentose phosphate pathway (PPP) [26]. Similar to how the population of cancer stem cells (CSCs) grows in response to chemotherapy, ionizing radiation, or oxidative stress [100,101,102]. It has been shown that increased oxidative PPP activity coexists with the acquisition of drug resistance, a characteristic of CSCs, in a number of cancer cell lines [103]. 

## 5. Significant Role of ROS Moieties in Cancer Cell Signaling Pathways 

ROS act as key regulators of several cellular processes including cell differentiation, proliferation, as well as death pathways, and therefore maintain ROS level balance via regulating the “REDOX” homeostasis process [63]. A controlled improvement in ROS levels within the bounds of “REDOX” homeostasis may provide a positive signal for H_2_O_2_ directed oxidation pathway of protein molecules, principally cystine residues, thereby initiating the cellular mechanism as the proliferation of cells [104]. The “REDOX” homeostasis, on the contrary, disruption leads to an overload of ROS, for example, irreversible oxidative DNA damage, leading to the death of cells. In recent times, well-established data showed that metabolically transformed with rapidly proliferating cancer cells produce a huge amount of ROS than neighboring normal cells, attempting to put cancer cells at a higher risk of exceeding the ROS threshold for apoptosis. This evidence suggests that further ROS production in uncontrolled growth cells (cancerous cells) can be used as a strategic plan to stimulate the death of cancer cells properly [86]. 

It also takes part in crucial roles in initiating, promoting, and advancing tumor progression [3], and activates oncogenes, namely Ras and c-Myc, at levels below the ROS threshold, which leads to p53-related DNA reparation and subsequently survival in cancer cells [105,106]. Additionally, it regulates these cellular processes via a great number of signaling pathways, and the data are represented in Figure 3, including mitogen-activated protein kinase (MAPK), phosphoinositide-3-kinase (PI-3K), extracellular regulated kinase (ERK), protein kinase B (PKB), and nuclear factor-kappa B (NF-κB), an inhibitor of kappa B kinase (IKK) [107,108]. For instance, ROS-dependent ERK activation regulates proinflammatory gene expression by phosphorylation of transcription factors [108,109]. Besides, anti-apoptotic proteins mainly BCL2 as well as BCL-XL are phosphorylated and downregulated when JNK is activated by ROS [110]. In response to ROS, PKB is phosphorylated by the IKK and then ubiquitinated, causing NF-κB to be activated and transported into the cellular nucleus. It stimulates the expression of anti-apoptotic genes [111]. Subsequently, ROS activate PI-3K directly, afterward converting phosphatidylinositol 4,5-bisphosphate (PIP2) into phosphatidylinositol 3,4,5-triphosphate (PIP-3) and resulting in transcriptional inhibition of the target genes of ACTs, glycogen synthase 3 (GSK 3) (mTOR 1) [112]. The ROS-mediated apoptosis process can be triggered using both intrinsic mitochondrial signaling pathways, and the extrinsic apoptotic death receptors-dependent signaling pathways. Increased ROS production depolarizes the mitochondria membrane, allowing for the escape of cytochrome c, and subsequently leading to the activation of the caspase-9 pathway, promoting the nucleotide bond to APAF-1, and consequently activating the caspase 3 pathways [113]. The development of different channels with mitochondrial membrane permeability also contributes to anti-apoptosis (i.e., BCL-XL or BCL-2), and pro-apoptosis (i.e., BAD, BAX, and BAK) proteins [114]. However, activation of both death receptors, as well as caspase 8-related pathways, has also increased ROS levels. Furthermore, ROS influences Fas- and/or TRAIL-dependent apoptosis, upregulating the p53-related death receptor. Hence, p53 controls cell death via regulating the anti-apoptotic (e.g., PUMA and NOXA) expression [115,116]. ROS also increases Ca^2+^-induced mitochondrial permeability transition pore opening, which promotes apoptosis [117]. 

## 6. Quercetin Upregulate the ROS Levels in Cancerous Cells 

As far as it is concerned, QC can affect the formation of ROS and ultimately promote apoptosis; it significantly enhances ROS levels within the cell as QC radicals (QC-O•) can form to scavenge reactive peroxyl radicals after peroxidase catalyzed oxidation [48]. For this purpose, QC could produce sufficient ROS which free radically-induced apoptosis at least by the signaling pathways including p38/ASK1/AMPKα1/COX2/AMPKα1 [118]. Thus, ROS generation proceeds to cause p38 and caspase-dependent activity, as well as activation of AMPKα and ASK1, which follow p38 recruitment and colocalization with protein kinase [119,120] and the COX-2 is a further downstream AMPKα1-mediate apoptosis-induced target [121]. 

### 6.1. Balanced Oxidative Stress in Cancerous Cells 

Diverse experimental work discovered that overexpression of endogenous antioxidants such as SOD is balanced in oxidative stress of various cancer cells because seven transformed cancer cell lines namely melanoma, colon cancer, breast, and ovarian cancer, neuroblastoma, pancreatic cancer, and leukemia produce more H_2_O_2_ free radicals and lymphocytic cells through blood samples from leukemia patients produce excessive intrinsic pro-oxidant such as free radicals as well as liver cancer cells produce more superoxide free radicals rather than non-transformed normal cells according to in vitro and in vivo experiments [122]. Recently, an in vitro experiment showed that a DNA repair molecule hMTHI and 8-oxo-dG accumulation were produced excessively in brain tumors as well as high-grade glioma that increased oxidative stress for responsible for tumor propagation, and these DNA reactive oxygen species have been evaluated by HPLC and ELISA test, stomach adenocarcinoma and esophagus squamous cell carcinomas produce more free radicals that induce more MnSOD enzyme for the body’s homeostasis as well as colorectal adenocarcinomas and liver metastases that produce more ROS and induce superoxide dismutase enzyme which has been detected by Western blot analysis and immunohistochemical staining, where SOD, glutathione S-transferase (GSH), and glutathione peroxidase (GPX) have been exchanged greater than normal non-transformed cells [122]. Pro-oxidants are balanced which are produced in transformed cells by changing the oncogenic transformation, metabolic activity, and decreasing the activity of P^53^ tumor suppressor gene in which the cancer cells need energy, DNA, RNA, lipids, proteins, and amino acids for progression of cancer cells growth, and this process occurs in mitochondria but in cancer cells there remains hypoxic or anoxic conditions for which ATP is produced by glycolysis rather than oxidative phosphorylation or the electron transport system, as well as the level of H+ ATP synthase β-subunit (β-F1-ATPase) which is decreased in tumor cells than the level of the β-F1-ATPase of non-transformed normal cells for which cancer cells uptake their energy from glucose in glycolysis via the EMP pathway and transcription factors named hypoxia inducible factors (HIFs) causing angiogenesis that moderates many pathways such as metabolic modification which is essential for cancer proliferation where oxygen is depleted and hypoxia-inducible factor regulates different genes which induce glucose transport for instant-GLUT 1/3 with metabolism including hexokinase 2 and pyruvate dehydrogenase kinase 1; as a result, ROS are more produced and induce HIFs signaling pathways, and ROS help in the initiation with proliferation of cancer [8]. At the same time, increasing ROS, the transformation of oncogenes in cancer cells was observed in a mouse xenograft model where the oncogene RAS version was overexpressed and transformed H-RASV which transformed BCR-ABL cell lines and more ROS were produced compared to non-transformed cell. As a result, tumor proliferation occurred and cancer cells survived but chemopreventers produced excessive ROS that caused cancer cell death as shown in Figure 4 [123]. 

Tumor suppressor gene p53 is decreased in cancer cells which upregulates antioxidants genes such as MnSOD2, tumor protein P^53^-inducible nuclear protein 1 (TP^53^INP 1), GPX, TP^53^-induced glycolysis and apoptosis regulator (TIGAR), and SESN 1/2, the sestrins, which encode antioxidant modulators of PRDXs for which the ROS levels are produced more in cancer cells but endogenous antioxidants balance the ROS levels [8]. 

### 6.2. Unbalanced Reactive Oxygen Species in Cancerous Cells by Quercetin

QC is a nutritional supplement derived from daily food and plants such as green tea, vegetables, and fruits that belongs to the flavonoid’s family. The pharmacological and bioactive function of quercetin is highly absorbed in the conjugated form which is metabolized into QC-3-glucuronidated, methylated, and QC-3′-sulfated in the intestinal tract in the form of glycosides by glycosidase enzyme and excreted with urine in the kidney as shown in Figure 5 [67]. 

Quercetin has anti-inflammatory activity that pulls out the nitric oxide, catalase, and cytokines, specifically TNF-α, IL-β, and IL-6, which are inflammatory mediators, and in the promoter region of pro-inflammatory genes which is transcribed by NF-κB transcription factor where the TNF-α induced is collected and the quercetin inhibits the expression of the genes and interrupts the lipid peroxidation or poly-unsaturated fatty acid that increases cancer propagation by prohibiting the enzyme lipoxygenase [124]. Human body cells have many defense mechanisms such as endogenous antioxidants such as SOD, PRX, GPX, CAT, glutaredoxins, and so on, detoxifying enzymes, non-enzymatic scavenger molecules (e.g., glutathione) protect the body cell from damage and are involved in repairing mechanisms [8]. It acts as an unstable free-radical scavenger named antioxidant which contains numerous hydroxyl groups and π orbitals for which quercetin is an electron and hydrogen donor and oxidized to form semiquinone and H_2_O_2_ reacts with •O_2_^–^ and further reacts with H_2_O_2_ by the enzyme of peroxidases, thus protecting cell degradation by removing these free radicals from the cells and producing the first oxidation component semiquinone which is unsteady, so this semiquinone oxidation reaction occurs to form quercetin-quinone [125]. Glutathione is a tripeptide and an important biomolecule that serves as a redox buffer regulating the REDOX state inside the cell, and glutathione is beneficial for immune system defenses and also important for cell repair against cell damage to protect the body, so it has antioxidant activity where QC-quinone interacts with the reduced GSH to form stable oxidized GSH such as 6-glutathionyl-QC and 8-glutathionyl-QC, and these glutathionyl-quercetin reversibly segregate to quercetin-quinone and reduced glutathione, then this oxidized quercetin reacts with reduced glutathione to form oxidized glutathione [126]. Reviewing several experiments using the model of rat liver nucleic extract, in high concentrations of oxidized GSH, unstable and toxic semiquinone and quinone derivatives of quercetin react with reduced GSH to form oxidized S-nitroso-glutathione by enhancing the nitric oxide levels for depleting GSH content to reduce cytotoxicity from the body’s cells that causes the diminution of membrane (mitochondrial) efficiency, phosphatidyl serine exposure, and mitigation of mitochondrial dependence to increasing the permeability of propidium iodide and DNA damage, along with quercetin combined with arsenic trioxide in U937 cells that cause the depreciation of protein resulting in apoptosis [8]. Conversely, low concentrations of reduced GSH content, semiquinone, and quinone derivatives of quercetin react with thiols groups of protein and cause cell death; ultimately, as a result, low concentration and high concentration of GSH content of QC causes cell damage and leads to cell death which inhibits cell proliferation in cancerous cells to increasing levels of ROS, notably semiquinone and quinone, which are highly unstable and toxic for cell proliferation and for which QC takes action as a pro-oxidant than antioxidant as shown in Figure 6 [118]. 

### 6.3. Quercetin-Mediated Apoptosis via Regulating Cancer Cell Signaling Pathways

Several in vitro results showed that the downregulation of heterogeneous ribonucleoprotein A1 (hnRNPA 1) expression by quercetin which is susceptible to apoptosis in PC-3 prostate cancer cells along with QC which induced pro-oxidants such ROS in hepatocellular cancer, and is converted to lipid to poly-unsaturated fatty acid (PUFA) which causes membrane damage, for which flavonoids such as quercetin have synergistic effects in cancer cells rather than synthetic drugs because they have many adverse effects that have been shown in human and animal trials [127]. The combination of quercetin and paclitaxel enhanced the anticancer therapeutic effect in the PC-3 cell line (prostate cancer), and decreased the adverse effects of paclitaxel to decrease its dose, so quercetin and PTX combined therapy has been upregulated in CHOP and GRP78, has cleaved caspase-3, and downregulated hnRNPA 1 to increase the arrest of different phases of the cell cycle, mainly G2/M and induced ER stress. Quercetin has a pro-oxidant effect rather than an antioxidant effect, so QC produced more ROS in PC-3 cell lines thus increasing ER stress, and G2/M phase arrest was evaluated using RT-PCR and Western blotting, and this combined therapy of QC+PTX caused PC-3 prostate cancer cell death [127].

Recently, numerous scientific reports state that QC and its analogs QC-semiquinone act as free radicals as they are more effective in arresting the S-phase and synthetic quercetin semiquinone arrested the G2/M phase of LoVo cells (human colon cancer), MCF-7 cells (human breast cancer), as well as QC inhibit the proliferation of G1 phase cell cycle of gastric cancer and it also has been increased the G2/M phase in HL-60 cell lines, so these reports showed that QC and its synthetic derivative quercetin semiquinone blocked the progression of S-phase cell cycle of cancers; thus quercetin and its analogs quercetin semiquinone had an anticancer proliferation effect in cancer cells and caused cancer cell death because quercetin and quercetin semiquinone increased ROS in cancer cells than normal cells [128]. Different in vitro research outcomes showed that QC downregulates the strong caspase inhibitor surviving in the HepG2, glioma, SW480 colon carcinoma, and non-small cell lung cancer cell lines by downregulation of putative mRNA which degraded the proteasome that causes cancer cell death. Quercetin regulated the expression of Hsp-90 heat shock protein which was downregulated in a caspase mechanism and the quercetin analog phenylisocyanate (PHICNQ) downregulated the Hsp-70 in K562 cell lines (human chronic myeloid leukemia) as a result causing apoptosis of cancer. In addition, ROS in cancer cells induced the ROS/AMPKα1/ASK1/P38 and the AMPKα1/COX2 pathways and stopped the P13K/Akt pathway which is responsible for cancer cell progression. Moreover, several studies showed that QC upregulated the death receptor (DR) induced by TNF-related apoptosis-inducing ligand (TRAIL) and downregulated the inhibition of caspase-8 named c-FLIP in prostate cancer, and hepatoma cells. In consequence, cancer cells are more sensitive to quercetin and normal cells are less sensitive to quercetin since ROS are more produced in cancer cells by quercetin [8].

## 7. ROS Are Responsible for Quercetin-Mediated Cell Death in Cancer 

QC activates apoptosis by the ROS-mediated mitochondrial pathway, resulting in membrane damage to mitochondria; consequently, cytochrome c is released from the mitochondria and exported to the cytosol, caspase-3, and caspase-7 pathways which are mainly activated [49,57]. QC can also influence ROS metabolism by reducing the GSH pool within the intracellular level effectively [129]. Due to long-term QC exposure, the monoblastic and lymphocytic U937 cells exhibited decreased H_2_O_2_ and enhanced GSH content, demonstrating the need for a reduced ROS expression level [57]. Toxic (reactive) oxygenation products, such as semiquinone and quinone, can be formed in the presence of ROS because these prefer to react with sulfate groups. Therefore, they are often found in higher concentrations than GSH [130,131]. Herein, QC reduces GSH concentration dependently; the higher the level of QC concentration, the larger the amount of GSH is exhausted, presumably due to GSH reacting with quinone, as well as semiquinone radicals derived from QC. GST (glutathione S-transferase) with nuclear GSH content decreases due to QC dose-dependent actions discovered within the isolated rat liver nuclei [132]. Specifically, during QC-mediated cellular apoptosis, the potential damage to the mitochondrial membrane, exposure to phosphatidylserine, and the reduction in mitochondrial mass are all early events that precede permeability to propidium iodide and DNA loss [133,134]. The effectiveness of QC-As_2_O_3_ (arsenic trioxide) combination therapy is also responsible for GSH depletion and the treatment applied within the U937 cells [52]. It has been reported that As_2_O_3_ detoxification is dependent mainly on GST enzymes and that As_2_O_3_ is highly reactive with thiol groups; of late, intracellular GSH depletion may elevate the intracellular free As_2_O_3_ concentration levels, which potentially enhance protein damage [135].

## 8. Molecular Mechanism for Quercetin-Induced Cancer and Cancer Stem Cell Death 

Due to its antioxidant properties, quercetin is used therapeutically for a variety of disorders. Mainly, the scavenging activity of ROS is converted to hydroxyl ions (OH^−^) and holds the electron exchange capability [136]. QC is lipophilic in nature and can easily pass through the cell membranes and latterly activate numerous intracellular signaling pathways, which are health effective [137]. It has been demonstrated that it mediates not only intrinsic (mainly mitochondrial) but also extrinsic (Fas/FasL) apoptotic cancer cell death (Figure 7) [57,138]. In addition to apoptosis, previous research has suggested that QC plays an important role in arresting the cell cycle and controlling the expression of CDK enzyme (cyclin-dependent kinases) [139]. Recent reports mention that it hinders the activity of cytochrome P450 (CYP) enzymes in hepatocytes and that most of them have crucial roles in the metabolism of drugs [140]. QC has also been shown to suppress metastatic protein expression such as MMPs (matrix metalloproteases) [50]. Actually, the metastasis process is also supported via neo-angiogenesis, and interestingly, QC is also noted to hinder neovascularization within the microenvironment of tumors [141]. Another antitumor property of QC is its ability to inhibit inflammatory mediators including IFN-γ, IL-6, COX-2, IL-8, iNOS, TNF-α, and many other cancer inflammatory mechanisms [142]. Investigating the mechanistic basis for these bioactive metabolites’ actions will aid in our understanding of cancer biology (Table 1). Besides, here is also included an extensive number of scientific outcomes, where QC is a vital therapeutic agent in cancer stem cells and mitigates the devastating role of cancers, and the results are represented in Table 2. 

### 8.1. ROS-Mediated Regulation of the MAPK/ERK1/2 Pathways 

Quercetin plays a crucial role in several signal transduction pathways. Herein, it directly activates the MAPK/ERK-mediated pathways, leading to the apoptosis process and the mechanism discovered within the A549 cell lines of lung carcinoma [69]. Besides, it induces the production of inflammatory and/or proinflammatory cytokines, and inhibits the formation of cyto-kinase LPS, which results in the suppression of iNOS by following the ERK, MAPK, as well as p38 mechanism [143]. Basically, ERK1 or 2 is the major section of the MAPK cascade mechanism, these build up kinase family proteins namely MEK, Raf, and ERK1/2 that work successively [144]. The active ERK 1 or 2 causes are reprogramming events linked to the expression of genes through the active phosphorylation process of various intracellular molecular target proteins and other transcription factors [75].

**Table 1 ijms-23-11746-t001:** Tabular representation of the anticancer potential of quercetin in different cancer types.

Cancer Type	Research/Experiment Type	Research Models/Cell Lines	Mechanism of Action	Outcomes	Reference
**Leukemia**	In vitro	U937 cell	Cell cycle arrest at G2/M, decrease in cyclin D, cyclin E and E2F, increase in the level of the cyclin B	Apoptosis and growth inhibition in The human leukemia cells	[49]
**Leukemia**	In vitro and In vivo	HL-60 AML cells	Induced caspase-8, caspase-9, and caspase-3 activation, PARP cleavage, mitochondrial membrane depolarization, induced intratumoral oxidative stress	Anticancer effects in acute myeloid leukemia (AML) cells	[145]
**Leukemia**	In vitro	Acute leukemia cell line, HL-60	Induces apoptosis in a caspase-3-dependent pathway by inhibiting Cox-2 expression and regulates the expression of downstream apoptotic components, including Bcl-2 and Bax	Inhibited cell proliferation and induced apoptosis in a time- and dose-dependent manner	[146]
**Liver cancer**	In vivo	Hyperplastic nodules in rat liver	Prevented DEN-mediated development of hepatocarcinoma and oxidative damage in rat liver	Potent therapeutic formulation against DEN-induced hepatocarcinoma	[147]
**Liver cancer**	In vivo	HepG2 cells	Induced apoptosis, alter cell cycle in hepg2 cells, decreased the gene expression of cyclin D1	Significantly inhibit the growth and proliferation of liver cancer cell.	[148]
**Colorectal cancer**	In vivo	ApcMin mice, and HCT116 tumors	Decreased tumor proliferation and development, increased apoptosis and p53 expression	Chemical modification of quercetin generates safe and efficacious agents for colorectal Cancer	[149]
**Colon cancer**	In vitro	HT-29 colon cancer cells	Induced caspase-3 cleavage, increased PARP cleavage, decreased the expression of Sp1, Sp3, Sp4 mrna, and survivin, decreased microrna-27a, and induced ZBTB10	Cytotoxic effects in colon cancer cells, Resulting in apoptosis.	[150]
**Colon cancer**	In vitro	CX-1, SW480, HT-29, HCT116	Downregulation of transcriptional activity of β-catenin/Tcf signal pathway, and cyclin D1 and the survivin gene	Inhibited proliferation in colon cancer cells	[151]
**Colon cancer**	In vivo	Male F344 rats	Decreased β-catenin accumulation in BCA-C; decreased number of ACF	Suppressed tumor growth and at high dose reduced colorectal carcinogenesis	[152]
**Lung cancer**	In vitro	H460, A549	Induction of DR5 and suppression of survivin expression	TRAIL-induced cytotoxicity in lung cancer cells	[153]
**Lung cancer**	In vitro	H460	Increased the expression of TRAILR, caspase-10, DFF45, TNFR 1, FAS, and decreased the expression of NF-κb, ikkα	Useful in the prevention and therapy of NSCLC	[154]
**Lung cancer**	In vitro	Human A549 lung cancer cells	Downregulation of the expression of cdk1 and cyclin B, increased PPAR-γ expression	Inhibiton of human A549 lung cancer cell growth	[155]
**Ovarian cancer**	In vitro and in vivo	A2780S ovarian cancer cells	Activated caspase-3 and caspase-9. MCL-1 downregulation, Bcl-2 downregulation, Bax upregulation, inhibited angiogenesis in vivo	Novel nano-formulation of quercetin with a potential clinical application in ovarian cancer therapy	[156]
**Ovarian cancer**	In vitro	SKOV3	Reduction in cyclin D1 level	Inhibited cell growth in ovarian carcinoma	[157]
**Breast cancer**	In vitro	MCF-7, HCC1937, SK-Br3, 4T1, MDA-MB-231	Decreased Bcl-2 expression, increasedBax expression, inhibition of PI3K-Akt pathway	Decreases proliferationand increases apoptosis in MCF-7 human breast cancer cells	[158]
**Breast cancer**	In vitro	MDA-MB-231	Induced the expression of E-cadherin and downregulated vimentin levels, modulation of β-catenin target genes such as cyclin D1 and c-Myc	Inhibited TNBC metastasis and also improve the therapeutic efficacy of existing chemotherapeutic drug	[159]
**Breast cancer**	In vitro	MCF-7	Suppressed the epithelial–mesenchymal transition process, upregulated E-cadherin expression, downregulated vimentin and MMP-2 expression, decreased Notch1 expression and induced PI3K and Akt phosphorylation	Potential therapeutic for the treatment of triple negative and hormone-sensitive breast cancer	[160]
**Breast cancer**	In vivo	MCF-7/DO X	Overcoming the drug efflux by ABC transporters and promoting PCD with the arrest of cell cycle, counteracted P-gp and BCRPPumps	Reverses multidrug resistance and restores chemosensitivity to human breast cancer cells	[161]
**Gastric cancer**	In vitro	GCSCs	Activation of caspase-3 and -9, downregulation of Bcl-2, upregulation of Bax and cytochrome c (Cyt-c)	Potential agent for the treatment of gastric cancer.	[162]
**Pancreatic cancer**	In vivo	PANC-1, PATU-8988	Decreased the secretion of MMP and MMP7, blocked the STAT3 signaling pathway	New therapeutic strategy for the treatment of pancreatic cancer cells That targets emt, invasion, and metastasis.	[163]
**Prostate cancer**	In vitro and in vivo	PC-3, HUVECs	Reduced angiogenesis, increased TSP-1 protein and mrna expression	Good foundation for applying quercetin to clinical for human prostate cancer in the near future	[164]

**Table 2 ijms-23-11746-t002:** Tabular representation of quercetin-mediated cancer stem cell killing with their mechanistic illustration.

Sort of Cancer	Research/Experiment Type	Specific Cell Line/s	Core Molecular Mechanism	Doses	Final Outcomes	Reference
**Colorectal cancer**	In vitro	HT29 cells	Induced G2/M arrest	75 µM	Enhanced the efficacy of low concentration of doxorubicin chemotherapy in inhibiting cell proliferation, enhance cytotoxicity and apoptosis	[165]
**Breast cancer**	In vitro	MDA-MB-231	Lowered the expression levels of proteins such as aldehyde dehydrogenase 1A1, C-X-C chemokine receptor type 4, mucin 1 and epithelial cell adhesion molecules responsible for tumorigenesis	50 μM	Suppressed breast cancer stem cell proliferation, self-renewal, and invasiveness	[166]
**Prostate cancer**	In vitro	PC-3 and LNCaP cells	Activated capase-3/7 and inhibit the expression of Bcl-2, surviving and XIAP in CSCs. Furthermore, inhibits epithelial-mesenchymal transition by inhibiting the expression of vimentin, slug, snail and nuclear β-catenin, and the activity of LEF-1/TCF responsive reporter	20 μM	Quercetin synergized with epigallocatechin gallate inhibited the self-renewal properties of prostate CSCs, inducing apoptosis, and blocking CSC’s migration and invasion	[167]
**Breast cancer**	In vitro	MCF-7 and MCF-7/dox cell lines	Downregulation of P-gp expression and eliminate BCSCs mediated by YB-1 nuclear translocation	0.7 μm	Enhanced the antitumor activity of doxorubicin, paclitaxel and vincristine by reversing multidrug resistance	[168]
**Prostate cancer**	In vitro	PC3, LNCaP and ARPE-19 cells	Down-regulated the expression of PI3K/PTEN, MAPK and NF-κB signaling pathways	40 μM	Quercetin inhibited PC3 and CD44+/CD133+ stem cell proliferation in a time- and dose-dependent manner.	[169]
**Pancreatic cancer**	In vitro	Human pancreatic CSCs (CD133þ/CD44þ/CD24þ/ESAþ)	Inhibited the expression of Bcl-2 and XIAP and activate caspase-3, attenuate transcriptional activities of Gli and TCF/LEF	20μM	Epigallocatechin-3-gallate with quercetin had synergistic inhibitory effects on self-renewal capacity of pancreatic CSCs	[170]
**Pancreatic cancer**	In vitro	PANC-1	Affected IL-1b, TNF-α, vimentin, N-cadherin, and ACTA-2 expressions	10μM	Quercetin could prevent Epithelial Mesenchymal Transition by reducing expression of N-cadherin	[171]
**Breast cancer**	In vitro	MCF-7 and MDA-MB-231 cells	Suppressed EGFR signaling and inhibited PI3K/Akt/mTOR/GSK-3β	50μM(MCF-7), 100μM (MDA-MB-231)	Gold nanoparticles-conjugated quercetin reduce cell proliferation through induction of apoptosis of breast cancer cell	[172]
**Pancreatic cancer**	In vitro	IA Paca-2, BxPC3, AsPC-1, HPAC and PANC1	Silencing RAGE expression by suppressing the PI3K/AKT/mTOR axis		Quercetin increased gemcitabine drug chemosensitivity in pancreatic cancer cells	[173]
**Colon cancer**	In vitro	HT-29, SW-620, and Caco-2 cells	Redistributed the TRAIL receptors and other components of the DISC complex into lipid rafts, which facilitates the formation of the DISC and the downstream signaling pathway, contributing to Bax conformational changes, release of cytochrome c, and apoptosis.	30 μmol/L	Quercetin enhanced tumor necrosis factor-related apoptosis inducing ligand (TRAIL)-mediated apoptosis	[174]
**Breast cancer**	In vivo	NOD/SCID mice	Inhibited the overexpression of Hsp27 through the regulation of epithelial mesenchymal transition and nuclear factor-kB	(50, 25 or 12.5 μM)	Effectively suppressed the overexpression of Hsp27 and inhibit the breast cancer stem cells	[175]

### 8.2. ROS-Mediated Regulation of the p53 Pathway 

The p53 pathway has been found to modulate cellular stress responses and to be vital in the direction of apoptosis in cancer. It is directly associated with MDM2 because of the complexities of p53, and is regulated at low levels by the regular proteasomal breakdown in unstressed cells [176]. Similar to other cell signaling molecules, reactive oxygen species (ROS) function as an influencer in the activation of p53 through upstream signal transduction and also act as a downstream component to induce cell death (Figure 8) [106]. Quercetin, like other omnipresent bioactive plant flavonoids, has been demonstrated to reduce overall tumor growth and metastasis [177]. Quercetin inhibited tumor cell proliferation by concurrently stimulating two divergent regulatory networks, p53 and NF-κB [70]. An investigation reported that in A549 human lung cancer cells expressing wild-type p53, cell growth arrest and apoptosis were accelerated by quercetin considerably [71].

Another study found that while quercetin increased p53 phosphorylation, it did not increase p53 mRNA transcription. Quercetin increased p21 expression while suppressing cyclin D1 expression, favoring cell cycle arrest. In favor of apoptosis, the enhancement of the Bax/Bcl-2 ratio with this sort of therapy is also mediated by quercetin. Noticeably, the prevention of p53 mRNA degradation at the post-transcriptional stage occurred through quercetin-mediated signaling [178].

Lee, Yoon-Jin et al. showed that experiment on MM MSTO-211H cells, quercetin induced apoptotic cell death at low concentrations. According to the study, the up-regulation of p53 induced by quercetin can be visualized at both mRNA and protein levels without altering its ubiquitination as well as enhancing caspase-3/7 activity and procaspase-3 and PARP cleavages. The subset of cells in the G2/M phase that has been obstructed by the pan-caspase inhibitor escalated later in a quercetin concentration-dependent approach [179]. Another study found that quercetin interacts with the human glioblastoma A172 as well as LBC3 cell lines, confirming increased ROS production, unregulated SOD1 and SOD2 expression, ATP depletion, and CHOP8 overexpression, and as a result, apoptosis of these brain cancer cell lines. They also reported enhanced caspase-9 expression and activity, supporting a mitochondrial apoptosis pathway [180].

### 8.3. ROS-Mediated Regulation of the JAK/STAT and TRAIL Pathways

The JAK-STAT pathway is a well-characterized signaling pathway involved in the production of a variety of inflammatory and/or proinflammatory cytokines along with growth-controlling factors. The connection of ligand molecules to specific receptors leads to JAK mechanism activation that also improves the auto-phosphorylation process and activates the STAT signaling pathways [181]. Several research studies revealed that the active form of mast cells directly increases the formation of Th2 cells such as cytokines, and at the same time reduces the secretion of cytokines mainly Th1. Similarly, mast cells regulate the expression of the JAK/STAT gene, which stimulates IL-13 development in the Th2 cell line [182]. 

Quercetin can effectively inhibit the signaling pathways of JAK-STAT in various inflammatory conditions. Additionally, quercetin therapy of activated T-cells suppressed the phosphorylation of the TiK-2 (tyrosine kinase-2), JAK2, STAT-3,4 enzymes stimulated by interleukin-12, resulting in diminished T-cell proliferation, and Th1 variation [183]. As a result, quercetin’s anti-inflammatory and anti-apoptotic properties play an important role in cancer prevention by modulating the TLR-2 (toll-like receptor-2) and JAK-2/STAT-3 pathways and significantly inhibiting STAT-3 tyrosine phosphorylation within the inflammatory cells [72]. QC-mediated pretreatment of cholangiocarcinoma cells inhibited the JAK/STAT cascade pathway-mediated activation of iNOS and activation of the ICAM-1 (intercellular adhesion molecule-1). Additionally, QC inhibited the inflammatory cytokine IL-6 and interferon-alpha activation [184]. 

### 8.4. ROS-Mediated Regulation of the AMPKα1/ASK1/p38 Pathways

Understanding the mechanisms of apoptotic pathways can lead to the development of effective cancer preventive and treatment options. Particularly, outside stimuli and mitogen-activated protein (MAP) kinases are two well-known intracellular signaling pathways that possess effective signals of the apoptosis transducing channel named the kinase cascade. Apoptosis signal-regulating kinase (ASK-1) is a redox sensor that belongs to the ROS-sensitive MAP kinase kinases family [120]. The chemopreventive drug, quercetin, reported potent apoptosis induction in cancer cells mediated by ASK-1. Quercetin is responsible for free radical-induced cell death by the production of sufficient reactive oxygen species (ROS) though it has conventional anti-oxidation activity in the body. Through the apoptosis signal-regulating kinase (ASK)-1 and mitogen-activated protein kinase pathways, we studied the regulation mechanism of quercetin-induced apoptosis [118]. A therapeutic technique for cancer treatment is to induce apoptosis in cancer cells [185]. Quercetin (3,30,40,5,7-pentahydroxyflavone) has a broad-spectrum of biological effects, along-with cancer prevention and tumor growth suppression [186]. After scavenging the peroxyl radicals present, quercetin is transformed to quercetin-O-, a radical form that emerges to be one of the multiple pathways participating in the formation of ROS by quercetin [118]. Due to the generation of quercetin radicals (quercetin-O•) throughout its interaction with peroxyl radicals, quercetin also enhances intracellular reactive oxygen species (ROS), inducing free radical-induced apoptosis via the ROS/AMPK1/ASK1/p38 and AMPK1/COX2 signaling pathways [187]. AMPK1 and ASK1 are activated when the quantity of ROS produced increases in which AMPK1 and ASK1 activate p38 and boost the activity of several caspases [186]. In the current study, it was reported that the quercetin-mediated apoptosis process is regulated by the activation of AMP-activated protein kinase (AMPK)/p38 MAPK signaling pathways and enhanced expression of sestrin 2. According to the findings, quercetin triggered apoptosis via raising the expression of sestrin 2 and creating intracellular reactive oxygen species (ROS). The activation of the AMPK/p38 signaling pathway by quercetin reported the induction of apoptosis, which was dependent on sestrin 2. Nevertheless, silencing sestrin 2 with short-interfering RNA (siRNA) demonstrated that quercetin had no effect on AMPK or p38 phosphorylation inside the cells where sestrin 2 was silenced [185]. Another AMPK1 downstream target involved in Quercetin-induced apoptosis is COX-2 [8]. Quercetin radicals have also been reported to reduce the intracellular glutathione (GSH) pool in a concentration-mediated manner, as well as initiate apoptosis via mitochondrial depolarization. A research study demonstrated that quercetin exhibited interaction with the estrogen binding site type II [187]. Apoptosis signal-regulating kinase 1 (ASK1) is a mitogen-activated protein (MAP) kinase that stimulates both the MKK3/MKK6–p38 MAP and MKK4/MKK7–JNK kinase pathways and is a key signaling route in many forms of stress-induced apoptosis. ASK1 has been discovered to have an important role in apoptosis triggered by stress, particularly oxidative stress, and endoplasmic reticulum (ER) stress [188]. AMP-activated protein kinase (AMPK) appeared to be an important regulator of quercetin-mediated ASK1/p38 activation. When the activity of AMPKa1 was inhibited with a synthetic inhibitor or siRNA, quercetin-activated ASK1 was unable to promote p38 activity. As a result, it is thought that quercetin’s apoptotic effects are mediated by the ROS/AMPKa1/ASK1/p38 signaling pathway, and that AMPKa1 is required for ASK1-induced apoptosis. The alanine substitution of NR4A2’s oxidative stress-mediated phosphorylation sites reduces NR4A2’s cytoplasmic translocation and pro-necrosis activity, implying that p38-dependent phosphorylation of NR4A2 is a critical regulation to bestow the necrosis-inducing activity to NR4A2 [189]. These research outcomes imply that the ROS/AMPKa1/ASK1/p38 signaling pathway is involved in quercetin-induced apoptosis, and that AMPKa1 is a key regulator of ASK1 [118].

### 8.5. ROS-Mediated Regulation of the RAGE/PI3K/AKT/mTOR Axis

AKT signaling is required for the maintenance of normal physiological conditions. AKT signaling is frequently activated in cancer, which keeps the cellular microenvironment of tumors in a highly oxidative state, which is necessary for tumor development. As a result, antioxidants are hypothesized to possess anticancer effects through their ability to disrupt the tumor microenvironment. According to the findings of the study, quercetin (QC) prevented AKT signaling, which led to less cell survival, inflammation, and blood vessel growth in mice with lymphoma [190]. In human pancreatic adenocarcinoma cells, QC enhanced cell death and chemosensitivity via the regulation of RAGE/PI3K/AKT/mTOR axis [173]. 

Angiogenesis is a critical phase in cancer growth and spread because it allows the growing tumor to get oxygen and nutrients. In vitro, in vivo, and ex vivo antiangiogenic potential of quercetin reported that the progression of human prostate cancer was suppressed through the regulation of VEGFR-2-mediated Akt/mTOR/P70S6K signaling pathways [191]. Quercetin treatment dramatically reduced the overexpression of pro-fibrotic factors (IL-6, IL-8, COL-1, COL-3, and LC3) as well as the increase in pro-fibrotic signaling mediators (mTOR and AKT) generated by LPS in WI-38 cells. Thus, nasogastric injection of QC significantly reversed an elevation in profibrotic markers (VEGF, IL-6, TGF, COL-1, and COL-3) and fibrotic signaling mediators (mTOR and Akt) in the rabbit tracheal stenosis research model, while also inactivating ATG5 [192]. QC blocked the PI3K/AKT/mTOR and STAT3 signaling pathways in primary effusion lymphoma cells, which induced cell death and autophagy [193]. In addition, quercetin showed its anti-breast cancer effect via inhibiting the Akt/AMPK/mTOR signaling cascade [194]. Because it inhibited the PI3K/Akt/mTOR signaling pathway and has been proven to reduce breast cancer stem cells (CD44+/CD24), it was identified as a candidate for the treatment of breast cancer [195]. 

Q-6-C-b-D-glucopyranoside, a naturally occurring derivative of quercetin, demonstrated anti-prostate cancer action via blocking the Akt-mTOR pathway through the aryl hydrocarbon receptor [196]. The generation of human breast cancer stem cells was inhibited due to the inhibition of the Notch1 and PI3K/Akt signaling pathways by quercetin-3-methyl ether [197]. In addition, quercetin-3-methyl ether stopped the growth of cancer in the esophagus by blocking the Akt/mTOR/P70S6k and MAPK pathways, which are important for the growth of cancer [198]. P53, Akt/mTOR pathway, and cancer cell growth were all downregulated in breast cancer cells when the vanadium quercetin complex is used, which is also associated with many apoptosis events [199]. On the contrary, Burkitt’s lymphoma cells are significantly more susceptible to death because of quercetin’s induction of a drop in c-Myc expression in addition to PI3K/AKT/mTOR signaling [200]. Quercetin inhibited glycolysis via activating the Akt-mTOR pathway, resulting in the activation of autophagy, hence inhibiting the migration of breast cancer cells [201]. In glycolysis-mediated HCC cells, it suppressed the hexokinase 2 and the Akt-mTOR pathway, which are involved in glycolysis-dependent cell proliferation [202]. Quercetin has been shown to suppress the Akt-mTOR pathway and hypoxia-induced factor 1 signaling pathway in gastric cancer cells, resulting in preventative autophagy [203]. 

Furthermore, quercetin possessed strong synergistic activity in combination with anti-sense oligo gene therapy (suppressing snail gene expression), it significantly inhibited the progression of renal cell carcinoma Caki-2 by the regulation of Akt/mTOR/ERK1/2 signaling pathways [204]. Quercetin nanoparticles inhibited the AKT/ERK/caspase-3 signaling pathway to induce autophagy and apoptosis in human neuroglioma cell lines in vitro and in vivo research models [205]. In breast cancer cell lines (MCF-7 and MDA-MB-231), quercetin coupled with gold nanoparticles promoted apoptosis by inhibiting the EGFR/P13K/Akt-mediated pathway [172]. 

### 8.6. ROS-Mediated Regulation of the HMGB1 and NF-κB Pathways

The HMGB1 (high mobility group box 1) protein that is exposed to the outside environment has the potential to trigger the production of TNF (tumor necrosis factor), IL-1, and other inflammatory cytokines from monocytes [206]. QC enhances the suppression of HMGB1-induced TNF and IL-1 mRNA production, indicating that it transmits signals inside cells that modulate the action of proinflammatory cytokines [207]. In HMGB1-induced gene expression, the activation of MAPK is important for the release of inflammatory cytokines such as IL-1 and TNF from neutrophils, macrophages, and endothelial cells. The release of cytokines induced by HMGB1 partially interferes with MAPK pathways in which the p38 phosphorylation or C-Junction of N-terminal kinases with extracellular signals regulate kinases in macrophages in a time-dependent manner by HMGB1 or LPS. Here, such a bioactive compound significantly inhibits the activity of each kinase enzyme by maintaining the phosphorylation process [208]. Additionally, a recent study reported that QC can easily modulate the signaling proteins (i.e., Cox-2, NF-κB) that may lead to the apoptosis process, and furthermore, suppress anti-apoptotic proteins, e.g., Bcl-2 as well as Bcl-xL, up-regulating Bax with several pro-apoptotic proteins [209,210]. In addition to activating MAPK, the nuclear factor-κB (NF-κB) pathway also includes HMGB1-induced cellular expression, and NF-κB-dependent gene expression is of significant importance when exposing cytokine [211,212]. Through their interaction with the IκBα (members of the IκB family), subunits of NF-κB (p50 as well as p65) take place in cells as inactive trimers within the cell cytosol [213]. Interestingly, QC suppresses the degradation of IκBα factor and NF-κB p65 in a significant manner. As a result, following stimulation with the following p65, HMGB1 or LPS, the critical mediator of NF-κB factor will become available to be regulated nuclear NF-κB genes and the compound QC most effectively inhibits the nuclear localization process [75]. 

### 8.7. ROS-Mediated Regulation of the Nrf2-Induced Phase II Enzyme and Signaling Pathways

Carcinogenesis can be suppressed through inductions from Phase II enzymes, including the GST (glutathione S transferase), NAD(P)H/NO (quinone oxidoreductase), UDP-GLT (UDP-glucuronosyl transferases), and HO-1 (heme oxygenase-1) [214,215]. The genes of such enzymes bearing the ARE (antioxidant replication elements), which are strictly regulated by Nrf2 nuclear erythroid, and which, in turn, are associated by Keap-1 (Kelch-like CEH-associated protein-1), therefore, an Nrf2 represent, also encourage their degradation through the path of the ubicin-dependent proteasome [216,217,218]. When cells are treated with ARE-mediated stimulants such as QC, the complex formation of “Nrf2-Keap1” dissociates, allowing the Nrf2 factor to be translocated from the cytosol to the nuclear part of the cell. By binding to ARE and subsequently forms a heterodimer structure with other transcription factors (TFs), which simultaneously promotes the transcription of phase II enzyme genes. [219,220]. QC has also been demonstrated to stimulate ARE binding activity and regulate the mRNA expression level of NQO1, the total mechanism that occurred in the HepG2 cell line in a dose-dependent manner [220]. In addition, QC regulated the prohibition of Nrf-2 protein degradation and maintained the reduced amount of posttranslational products of Keap-1 proteins without any impairment of the complex structure dissociation like- “Keap-1-Nrf2” [220]. A cross-mediated increase in enzymes of phase II Caco-2 adenocarcinoma with duodenum adenocarcinoma of HuTu 80 cells in human CRAC (colorectal adenocarcinoma) was shown [221]. The effect of QC on nuclear translocation of Nrf-2 in a time-dependent manner, and increased expression level in HepG2, MgM (malignant mesothelioma) MSTO-211H, and H2452 cells at mRNA and protein quantity has been reported recently [222]. Moreover, the therapeutic efficacy of QC has also been defined in rat models through the activation of Nrf-2/HO-1 against high glucose-induced damage [223]. Until now, little is known about the precise biochemical pathways by which QC triggers Nof2-based gene expression. Nevertheless, evidence suggests that the improvement of diverse signal transduction cascades, along with the MAPK, modulates cross-induced interpretation of transcriptional genes (MAPK). The p38 and ERK-generated translocation of Nrf2 to nuclei between MAPK signal pathways were shown to be responsible for the successive initiation and activity of HO-1 expression [224,225,226]. Indeed, QC (50 µM) protected RAW264.7 macrophages from the induction of apoptosis by up-regulating phase II enzymes, which include HO-1, via ERK pathway-dependent mechanisms [225]. Similarly, cross-treated human hepatocytes were mentioned in MAPK HO-1-dependent upregulation and successive preservation against ethanol-induced oxidative damage [226]. 

## 9. Quercetin Induced Cell Death through Cell Cycle Arrest → G1, G2/M, etc. Phase

In apoptotic pathways, QC arrested several specific control points of the cell cycle. These cell cycles are controlled by the cyclins and Cdks (cyclin-dependent kinases) pathways. The Cdks are controlled positively and negatively by Cdkis (cyclin-dependents kinase inhibitors) [227,228,229]. The initiation of the G1 phase arrest by QC has already been exhibited in human vascular smooth muscle cells due to inhibition of the Cdk-4, cyclin D1, and Cdk2/E proteins, but activation of the p21 [230]. Likewise, QC was shown to enhance the activation of the Rb (retinoblastoma) gene, which was accompanied by a cell cycle arrest in CNE2 and HK1 cells under the respective phase: G0/G1 or G2/M [50]. In an in vitro study, its concentration-dependent impacts on cell enlargement suppression and increased cell cycle arrest have been further described [177]. Dose- and time-based QC treatment in SCC-9 cells activated the synthesis of DNA and TS (thymidylate symptoms), a critical S-phase enzyme and arrested site of the QC moiety [231]. Cdk-1 and cyclin B-1 enzymes were downregulated; and pRb protein phosphorylation followed by stops of the G1, G2, and M phase in the cell cycles, respectively, was found in various cancer cell lines through intermediate upregulation of p21, p27, p53, and Chk-2 [48,232]. Moreover, time- and dose-dependent QC therapy following the arrest in the S phase of MCF-7 cells following Cdk2, and cyclin A, B downregulation and upregulation of p53, p57 [158,233]. QC induces arrest of the G2 phase of cancerous cells with decreased cyclin D1, cyclin A, cyclin B, and Cdk-1 in adult male Wistar rats [234]. Additionally, QC induces the suppression of ovarian cancer proliferation at the cellular level (i.e., SKOV-3 cells), and also maintains the malignant tumor growth of MgM MSTO-211H and H2452 via suppressing the progression of the cell cycle from G0/G1 to G2/M phases and finally inducing cellular apoptosis [235,236]. Another study revealed that QC protects the leukemia cell lines, mainly U937, from entering the G-1 phase by inhibiting HSP27-induced cyclin D1 [237]. 

## 10. Concluding Remarks and Future Directions 

To sum up, here we demonstrated the quercetin effects on cancer therapy, and multiple in vitro as well as in vivo studies have demonstrated numerous mechanisms of ROS-mediated regulation for inhibiting different tumorigenesis signal transduction. QC has been shown to be safe with no documented side effects when used to treat human cancer. Given the numerous benefits associated with QC and its derivatives, it is high time to continue investigating these molecules’ effects on cancer prevention with treatment. Nevertheless, there is currently insufficient data regarding its precise mechanism of action to support its clinical implication in human cancer treatment. As a result, further research should focus on elucidating the precise mechanisms of action of QC. Likewise, further clinical trials of the efficacy with the bioavailability of QC for potential use in humans in biological systems are needed, in particular in cancer treatment. In addition, the conversion of QC into its metabolites should be examined while evaluating the effectiveness and pharmacokinetics of QC for any further pharmaceutical use. 

The combination of xenobiotics and QC affects the reactivity of QC; however, there are several metabolic byproducts of it in conjugation that demonstrate favorable bioactivity. It is essential to continue investigating the mechanisms of action of QC, particularly its ability to suppress carcinogenicity in rabbits. The findings of recent epidemiological research suggest the lack of enough evidence for supporting the use of QC in the prevention of human cancer. Additional epidemiological studies are needed to determine the role of QC in preventing cancer in humans. The update of the flavonoid database will provide critical knowledge for epidemiological research studies in the future. QC is limited due to its slight water solubility along with oral bioavailability for pharmaceuticals. To improve the solubility as well as bioavailability of QC in humans, numerous research approaches have already been employed, including new drug delivery systems such as nanostructures, liposomes, and so on. These and other strategies will aid in our understanding of QC’s full potential in cancer chemoprevention.

## Figures and Tables

**Figure 1 ijms-23-11746-f001:**
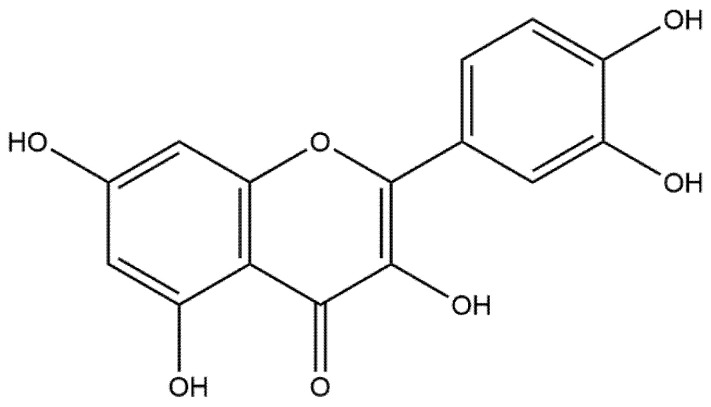
Chemical structure of quercetin.

**Figure 2 ijms-23-11746-f002:**
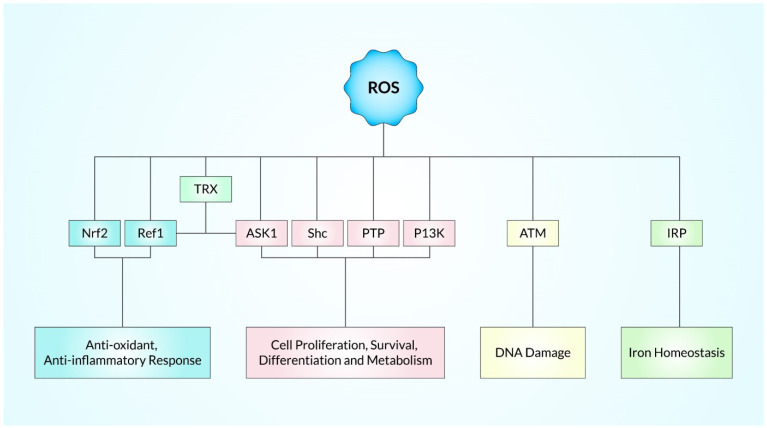
Both exogenous as well as endogenous regulatory sources of ROS along with enzymatic, and nonenzymatic antioxidants.

**Figure 3 ijms-23-11746-f003:**
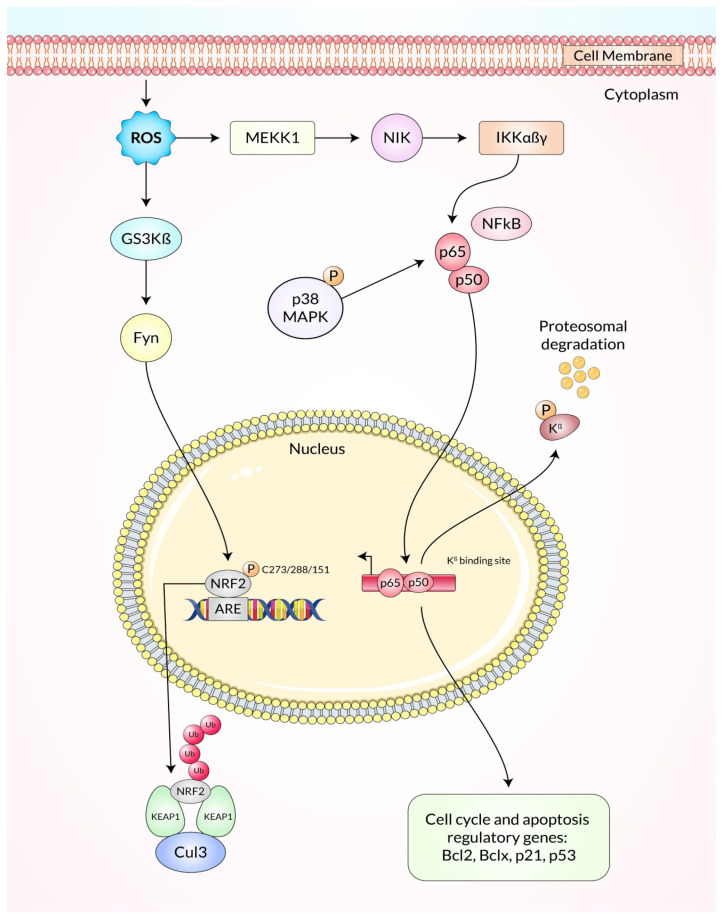
Intracellular cell signaling pathways mediated by ROS.

**Figure 4 ijms-23-11746-f004:**
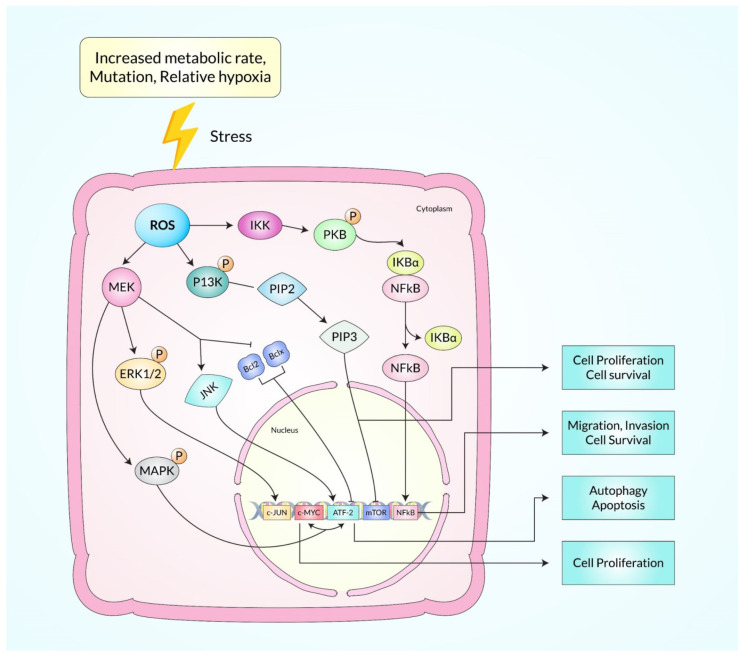
Diagrammatic illustration of oncogenic transformation and immortalized conditions by the presence of chemotherapeutic agents that augment ROS production.

**Figure 5 ijms-23-11746-f005:**
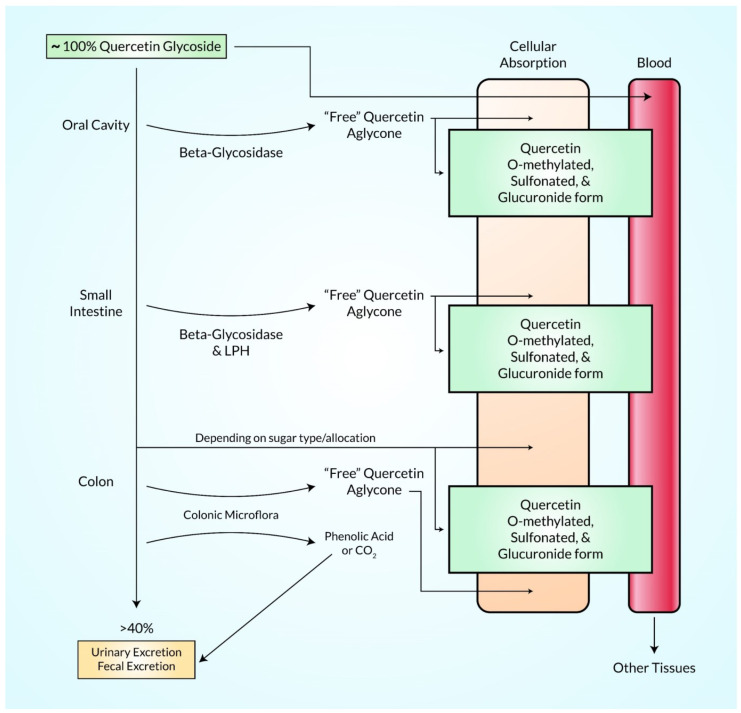
Illustration of possible pathways by which QC is absorbed, digested, metabolized, and excreted within body. Typically, QC glycoside is ingested orally, and partially digested in the oral cavity, surplus QC is then digested, and absorbed at multiple sites in GI tract. QC undergoes modification, and then enters the circulatory system in a conjugate form. The circulatory system delivers QC to other tissues in mostly conjugated forms, and once QC reaches the target tissues it can likely be converted back into the parental compound.

**Figure 6 ijms-23-11746-f006:**
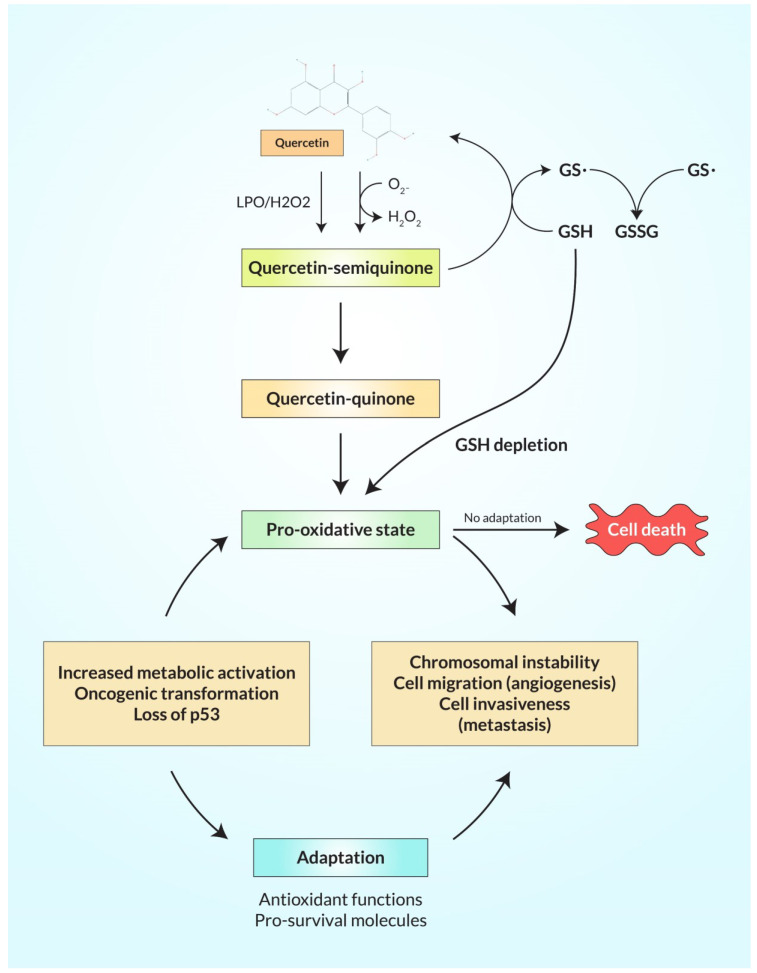
Diagrammatic pathway related to the effects of QC as both antioxidant, and pro-antioxidant in the presence of low and high levels of reduced GSH, both effects are strongly dependent upon the reduced GSH enzyme. During oxidative stress, in the presence of peroxidase, QC reacts with H2O2 to form a semiquinone radical that is rapidly oxidized to QQ, has a pro-oxidant effect, high reactivity towards protein thiols, and DNA leads to cell damage, and cytotoxicity. QQ preferentially reacts with GSH to stable protein oxidized QC adducts namely 6-GSQ, and 8-GSQ. In the presence of high GSH concentration, QQ reacts with GSH to form GSQ, and QQ cannot exert its cytotoxic effects, by contrast QQ reacts with protein thiols, and leading to cellular damage in the depletion of the GSH enzyme level.

**Figure 7 ijms-23-11746-f007:**
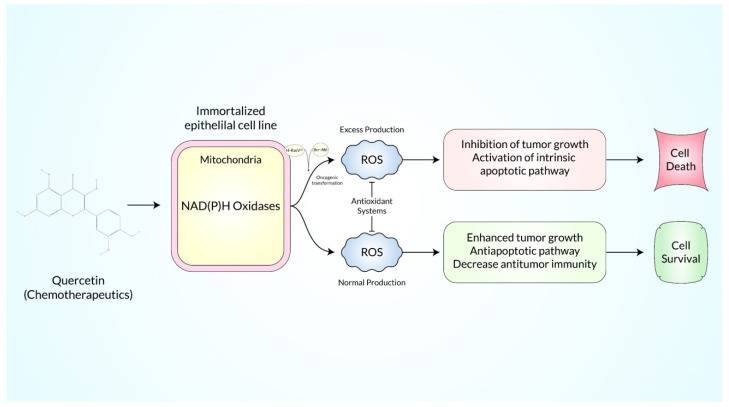
The anticancer pathways and mechanisms of action induced by quercetin.

**Figure 8 ijms-23-11746-f008:**
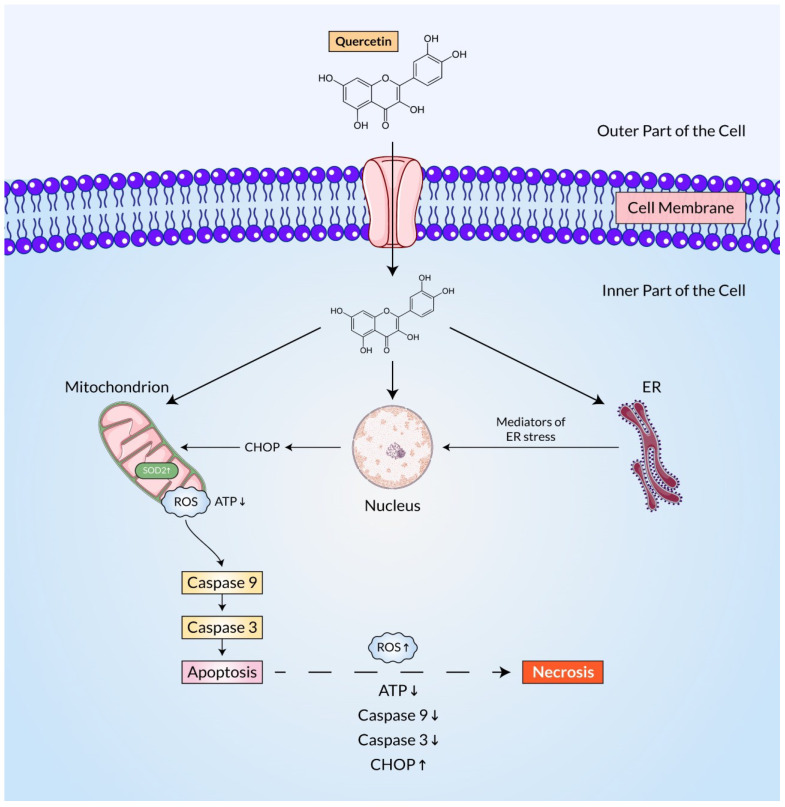
Mechanistic illustration of the function of quercetin in cancers cells by regulating ROS and p53-related pathways.

## Data Availability

The data presented in this study are available in this paper.

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
