# Peer review of "A Comprehensive Analysis and Anti-Cancer Activities of Quercetin in ROS-Mediated Cancer and Cancer Stem Cells"

_ijms, 2022, doi:10.3390/ijms231911746_

Round 1

Reviewer 1 Report

Comments:

No much literature was reviewed with regards to cancer stem cells throughout the paper. The authors may consider to remove the words accordingly. Similarly, no specific information was given on ROS production in cancer vs normal in part 4. This paragraph can combined with part 3 with a new subtitle.  

Author Response

Comment 01) Not much literature was reviewed with regards to cancer stem cells throughout the paper. The authors may consider to remove the words accordingly.

>>Response: In our updated manuscript, we have added more information about the cancer stem cells in the page 6, lines (242-258). Also, we have removed some unusual contents from our previous manuscript.

Comment 02) Similarly, no specific information was given on ROS production in cancer vs normal in part 4. This paragraph can combined with part 3 with a new subtitle.  

>>Response: In our revised manuscript, we have added more information in the Part 04 about the ROS production, we hope, it is OK now.  

Reviewer 2 Report

The review focuses on an important compound QC and its role on ROS-mediated carcinogenesis. The review is a very comprehensive work with almost 250 references. I like that it is very clearly written, with a useful Table and very clear, easily understandable figures. There is a nice overview on relevant aspects of oxidative stress, signal transduction pathways, etc. My favorite part is the pharmacokinetics summary (Fig. 5) with details on the conjugated forms of QC. It is hard to find major points for improvement of this manuscript, and therefore, I just mention a few minor suggestions below.

1) If applicable,  I would insert 1-2 figures on the actual atomic resolution molecular interaction of QC with its macromolecular (protein) target. This may attract the attention of structural biologist and help to understand how QC participates in the "hard links" in signaling or other pathways.

2) In Fig. 6, I would insert Lewis structures of the compounds involved, not just the names in boxes. This would help the understanding of the chemistry of the events mentioned there.

3) Text of Section 8 should be a bit more explanatory on Fig. 7. This  was a bit difficult to follow.

Author Response

Comment 01) The review focuses on an important compound QC and its role on ROS-mediated carcinogenesis. The review is a very comprehensive work with almost 250 references. I like that it is very clearly written, with a useful Table and very clear, easily understandable figures. There is a nice overview on relevant aspects of oxidative stress, signal transduction pathways, etc. My favorite part is the pharmacokinetics summary (Fig. 5) with details on the conjugated forms of QC. It is hard to find major points for improvement of this manuscript, and therefore, I just mention a few minor suggestions below.

>>Response: Thank You Sir for Your Valuable Appreciations of Our Works.

Comment 02) If applicable, I would insert 1-2 figures on the actual atomic resolution molecular interaction of QC with its macromolecular (protein) target. This may attract the attention of structural biologist and help to understand how QC participates in the "hard links" in signaling or other pathways.

>>Response: Dear Sir, according to your great suggestion, we have edited our Figure 2, 3, 4 and 7. Now in our updated manuscript, the structural biologists will easily find their necessary information.  

Comment 03) In Fig. 6, I would insert Lewis structures of the compounds involved, not just the names in boxes. This would help the understanding of the chemistry of the events mentioned there.

>>Response: Sir, in our updated manuscript, we have edited the figure 6 according to your suggestion.

Comment 04) Text of Section 8 should be a bit more explanatory on Fig. 7. This  was a bit difficult to follow.

>>Response: According to your suggestion, we have edited our figure 7 according to our information. Now the reader can easily understand.